# Protective effects of stem cells from human exfoliated deciduous teeth derived conditioned medium on osteoarthritic chondrocytes

**Suleiman Alhaji Muhammad**[1,2], **Norshariza Nordin**[3], **Paisal Hussin**[4], **Muhammad Zulfadli Mehat**[5], **Noor Hayaty Abu Kasim**[6,7], **Sharida Fakurazi**[1,5]*

1 Institute of Bioscience, Universiti Putra Malaysia, Selangor, Malaysia, 2 Department of Biochemistry, Usmanu Danfodiyo University, Sokoto, Nigeria, 3 Department of Biomedical Science, Universiti Putra Malaysia, Selangor, Malaysia, 4 Department of Orthopaedics, Columbia Asia Hospital, Selangor, Malaysia, 5 Department of Human Anatomy, Universiti Putra Malaysia, Selangor, Malaysia, 6 Dean's Office, Faculty of Dentistry, Universiti Kebangsaan Malaysia, Kuala Lumpur, Malaysia, 7 Faculty of Dental Medicine, Universitas Airlangga, Surabaya, Indonesia

* sharida@upm.edu.my

**Data Availability Statement:** All relevant data are within the manuscript.

## Abstract

Treatment of osteoarthritis (OA) is still a major clinical challenge due to the limited inherent healing capacity of cartilage. Recent studies utilising stem cells suggest that the therapeutic benefits of these cells are mediated through the paracrine mechanism of bioactive molecules. The present study evaluates the regenerative effect of stem cells from human exfoliated deciduous teeth (SHED) conditioned medium (CM) on OA chondrocytes. The CM was collected after the SHED were cultured in serum-free medium (SFM) for 48 or 72 h and the cells were characterised by the expression of MSC and pluripotency markers. Chondrocytes were stimulated with interleukin-1β and treated with the CM. Subsequently, the expression of aggrecan, collagen type 2 (COL 2), matrix metalloproteinase-13 (MMP-13), nuclear factor-kB (NF-kB) and the level of inflammatory and anti-inflammatory markers were evaluated. SHED expressed mesenchymal stromal cell surface proteins but were negative for haematopoietic markers. SHED also showed protein expression of NANOG, OCT4 and SOX2 with differential subcellular localisation. Treatment of OA chondrocytes with CM enhanced anti-inflammation compared to control cells treated with SFM. Furthermore, the expression of MMP-13 and NF-kB was significantly downregulated in stimulated chondrocytes incubated in CM. The study also revealed that CM increased the expression of aggrecan and COL 2 in OA chondrocytes compared to SFM control. Both CM regenerate extracellular matrix proteins and mitigate increased MMP-13 expression through inhibition of NF-kB in OA chondrocytes due to the presence of bioactive molecules. The study underscores the potential of CM for OA treatment.

**Funding:** This research work was supported by Putra Grant, Universiti Putra Malaysia with grant number: GPB/9657800.

**Competing interests:** The authors have declared that no competing interests exist.

## Introduction

Osteoarthritis (OA) is among the major cause of disability [1] and is most common in the age-ing population. However, with the increasing number of younger athletes presenting with symptoms of OA [2] call for an urgent search for less invasive treatment strategies for the disease. The work-related impacts of OA are workplace limitation and job loss due to disability [3] that have both short-and long-term socioeconomic burden.

OA has remained a major clinical challenge due to limited intrinsic healing capacity and inadequate cartilage regeneration by chondrocytes. Inflammatory mediators are actively involved in the degeneration process in OA [4]. These factors coupled with matrix-degrading enzymes cause a homeostatic shift from synthesis to degradation of the extracellular matrix, thereby aggravating the disease condition. Currently, there are no effective treatments to restore native tissue integrity of the damaged cartilage. The existing therapeutic strategies for OA are mostly effective in pain or symptom relief and offer short-term benefits.

Cell-based therapies such as autologous chondrocyte implantation (ACI) and mesenchymal stromal cell (MSC) transplantation have evolved as therapeutic strategies for OA [5–7]. However, ACI is associated with drawbacks of low chondrocyte yield during isolation and possibility of chondrocyte de-differentiation during *in vitro* expansion. Similarly, MSC therapy has its challenges, which includes paucity of standardised functional characterisation and this hinders large-scale production of MSCs with consistent biological activities for clinical trials [8]. However, both the results of preclinical and clinical studies have shown that MSCs are capable of cartilage repair and regeneration in OA [6, 9]. Early studies attributed the therapeutic benefits of MSCs to their differentiation capacity. Contrary to this assertion, recent studies showed that MSC exerts treatment effects due to the secretion of functional bioactive molecules [10, 11]. These biomolecules are capable of modulating tissue microenvironment that allows intercellular communication for tissue repair and regeneration. Pulp from human deciduous teeth is an ideal source of MSCs for the production of secretome because of the presence of cells with high proliferative capacity. Therefore, this study was designed to assess how conditioned medium (CM) from SHED could modulate tissue microenvironment of catabolic processes in OA chondrocytes.

## Materials and methods

### Chondrocyte isolation and culture

The Medical Research and Ethics Committee, Ministry of Health, Malaysia approved the protocol with reference number [KKM.NIHSEC/P17-1554 (5)]. The written informed consent was obtained from the patients (n = 10). Cartilage was obtained from OA patients undergoing total knee arthroplasty in Hospital Serdang, Malaysia. All experiments were conducted in compliance with current Good Clinical Practice standards and in accordance with relevant guidelines and regulations set forth under the Declaration of Helsinki. The isolation of chondrocytes was done as previously described [12]. Briefly, cartilage was minced into small pieces and digested overnight with 0.2% of collagenase II (275U/mg; Gibco, United States) in Hank's balanced salt solution (Gibco, United States). After isolation, the cells were pulled together for subsequent experiments. The chondrocytes were cultured in an incubator under standard conditions of 37˚C and 5% $CO_2$ throughout the experiment.

### Multilineage differentiation of SHED

SHED were isolated at the Department of Restorative Dentistry, Faculty of Dentistry, Universiti Malaya and were previously characterised [13]. To confirm the MSC identity of the SHED,

the cells were subjected to chondrogenic, adipogenic and osteogenic differentiation using StemPro differentiation medium (StemPro Gibco, United States). The cells were maintained at 37°C, and 5% $CO_2$. For chondrogenic differentiation, a micromass culture was prepared by seeding 20 μl droplet of cell suspension ($4\times10^6$ cells/ml) in the centre of a multi-well plate and incubated for 2 h to allow cell attachment. After incubation, the warm chondrogenic medium was added to the wells whereas, DMEM/F-12 (Gibco, United States) supplemented with 10% fetal bovine serum (FBS) (Gibco, United States) was maintained for the undifferentiated wells. After 21 days, the cells were washed, fixed in 4% paraformaldehyde (PFA) (Sigma-Aldrich, United States) for 30 min at room temperature and stained with 1% alcian blue (pH 2.5) (Nakalai Tesque, Japan) for 1 h.

The cells were seeded at a density of $4 \times 10^3$ cells/cm$^2$ in a 24 well plate for osteogenic differentiation. At 90% confluence, the culture medium was switched to osteogenic induction medium for the differentiated wells, whereas the control cells (undifferentiated) were maintained in DMEM/F-12 supplemented with 10% FBS. At the end of 21 days differentiation, the culture medium was removed and cells were washed with phosphate-buffered saline (PBS) (Gibco, United States). The cells were then fixed in 4% PFA for 30 min at room temperature, rinsed with PBS and stained with 40 mM Alizarin Red S (Nakalai Tesque, Japan) solution (pH 4.2) for 5 min.

SHED were seeded at a density of $1\times 10^4$ cells/cm$^2$ in 24 well plates for adipogenesis. At 90% confluence, the culture medium was changed to adipogenic differentiated medium, whereas the undifferentiated wells were maintained in DMEM/F-12 supplemented with 10% FBS. After 21 days of differentiation, the cells were washed with PBS and fixed in 4% PFA for 30 min at room temperature. The PFA was removed and the cells were rinsed with PBS. Then, 60% of isopropanol (Nakalai Tesque, Japan) was added and incubated for 5 min at room temperature. After aspirating isopropanol, the cells were stained with Oil Red O (Nakalai Tesque, Japan) solution (0.5% Oil Red O in isopropanol and the stock solution was further diluted 3:2 in distilled water) for 15 min. After staining, the cells were washed with distilled $H_2O$ for at least three times and images were captured using a Nikon inverted microscope (Nikon Eclipse TS100).

## Preparation of CM

Conditioned medium was prepared as previously described [14] with modifications. SHED (P6-8) were seeded at densities $5 \times 10^3$ cells/cm$^2$ in DMEM/F-12 supplemented with 10% FBS and incubated until 70% confluence. Then, the culture medium was removed and the cells were washed three times with PBS. After rinsing the cells, the medium was changed to serum-free DMEM/F-12 and incubated for 48 or 72 h. The medium was collected and centrifuged at 2500 rpm for 10 min. The collected CM was filtered using a 0.22 μm filter (Corning, United States) and stored at -80°C until needed. The CM collected after 48 and 72 h of incubation were designated as S48 and S72, respectively.

## Characterisation of MSC phenotype and pluripotent markers

After the collection of CM, the SHED were evaluated for MSC phenotype to confirm whether the cells had maintained their MSC identity during the preparation of CM. Human MSC analysis kit (BD Biosciences, United States) was used for the analysis according to the manufacturer's instructions. The cells were analysed using a flow cytometer (BD FACSCanto II). Data acquisition and analysis were done using FACSDiva version 6.1.3. The positive MSC surface proteins assessed were CD44, CD73, CD90 and CD105, whereas negative MSC phenotype were CD11b, CD19, CD34, CD45 and HLA-DR.

After collection of CM, the cells were washed with cold PBS for expression of pluripotent markers. The cells were fixed in 4% PFA for 20 min at room temperature. After incubation, PFA was removed and the cells were washed three times with cold PBS supplemented with Tween 20 (PBST). Subsequently, cells were permeabilised with PBS supplemented with 0.1% Triton X-100 and incubated at room temperature for 15 min, which was followed by washing with PBST. The cells were blocked using 1% bovine serum albumin (BSA) (Thermo Fisher Scientific, United States) supplemented with 0.1% Tween 20 for 30 min at room temperature. After washing with PBST, the cells were incubated with rabbit polyclonal antibodies to human NANOG, OCT4 and SOX2 (GeneTex, Taiwan) overnight at 4°C. Thereafter, the cells were washed with cold PBST and incubated with goat anti-rabbit IgG (H & L)-FITC (Invitrogen, United States) at room temperature for 2 h. The concentrations were 10 μg/mL each for NANOG and OCT4, and 25 μg/ml for SOX2, whereas goat anti-rabbit IgG (H + L)-FITC was 2 μg/ml. The cells were counterstained with DAPI (Nacalai Tesque, Japan) for 10 min and the images were captured using a Zeiss fluorescence microscope (Axio Vert A10).

## ELISA

The level of transforming growth factor β1 (TGF-β1), interleukin 10 (IL-10), and interleukin 6 (IL-6) were determined from the CM and culture medium from the treated OA chondrocytes using human Qantikine ELISA kits (R and D systems, United States). MMP-13 was assayed from OA chondrocytes treated with CM with ELISA kit (Elabscience, China). The assays were carried out according to the protocol of the manufacturers. Briefly, 100 or 200 μl of CM or culture medium was added to 96 well microplate wells coated with monoclonal antibody to the factor of interest and incubated for 1.5 or 2 h. The plate was washed and horseradish peroxidase-conjugated factor-specific antibody was added to each well and incubated for 1 or 2 h. Substrate solution was added after washing and incubated for another 30 minutes after which the reaction was terminated by adding the stop solution. The absorbance was measured at 450 nm with a microplate reader (Sunrise, Tecan).

## Chondrocyte stimulation and treatment

Chondrocytes were stimulated with 10 ng/ml of interleukin-1β (Gibco, United States) as previously described [15]. To assess the effect of CM on chondrocyte viability, cells were seeded at a density of $1 \times 10^4$ cells/cm$^2$ in 96 well plates (n = 4) and cultured for 48 h. At the end of 48 h, the medium was aspirated and the cells were incubated with DMEM/F-12 supplemented with 0.5% FBS for 12 h to allow the cells to synchronise. Subsequently, the cells were treated with CM in the presence of IL-1β for 24, 48 and 72 h, respectively. The assay was done using a cell counting kit 8 (CCK-8) (Nacalai Tesque, Japan). After treatment, 10 μl of CCK-8 reagent was added to each well and incubated for 3 h and the absorbance was measured at 450 nm with a microplate reader. The control groups were non-stimulated cells incubated in DMEM/F-12 supplemented with 10% FBS i.e. complete culture medium (CCM) and stimulated cells incubated in SFM or SFM supplemented with TGF-β1. Results were normalised to CCM control and expressed as a percentage of cell viability.

For the assay of TGF-β1, IL-6, IL-10 and MMP-13 after treatment, chondrocytes (n = 3) were stimulated and treated as earlier described. The cell supernatant was collected after 48 h of treatment for analysis using ELISA kits according to the manufacturer's instructions.

## qRT-PCR

The mRNA expression levels of aggrecan, collagen type 2 (COL 2), MMP-13 and NF-kB were evaluated after the treatment of OA chondrocytes with CM. Chondrocytes stimulation and

**Table 1. Primer sequence.**

| Gene | Gene name | Primer sequence 5' 3' | Reference |
|------|-----------|----------------------|-----------|
| ACAN | Human Aggrecan | F: TGA GGA GGG CTG GAA CAA GTA CC | [16] |
| | | R: GGA GGT GGT AAT TGC AGG GAA CA | |
| COL 2 | Human Collagen Type 2 | F: CCC TGA GTG GAA GAG TGG AG | [17] |
| | | R: GAG GCG TGA GGT CTT CTG TG | |
| MMP-13 | Human Matrix Metalloproteinase-13 | F: CTT AGA GGT GAC TGG CAA AC | [18] |
| | | R: GCC CAT CAA ATG GGT AGA AG | |
| NF-kB | Human Nuclear Factor-kappa B | F: ATG GCT TCT ATG AGG CTG AG | [19] |
| | | R: GTT GTT GTT GGT CTG GAT GC | |
| GADPH | Human Glyceraldehyde-3-Phosphate Dehydrogenase | F: CAG AAC ATC ATC CCT GCC TCT | [18] |
| | | R: GCT TGA CAA AGT GGT CGT TGA G | |

treatment were done as described under cell viability section. After 48 h of treatment, the cells were then washed with PBS and trypsinised with 0.05% trypsin-EDTA (Gibco, United States). The cell suspension was centrifuged at 1200 rpm for 5 min.

The pellet was then subjected to total RNA extraction with FavorPrep total RNA mini kit (Favorogen Biotech, Taiwan) according to the manufacturer's protocol as previously described [12]. A quantity of 100 ng of extracted RNA was used to synthesized cDNA with qPCRBIO cDNA synthesis kit (PCR Biosystems, United Kingdom) as previously reported [12]. The primer sequence is depicted in Table 1.

After optimisation of the primers, the data were collected and processed with BioRad CFX96 qPCR Detection System using 1 cycle at 95°C for 2 min to activate the polymerase, followed by 40 cycles at 95°C, 5 sec for denaturation and 60°C, 30 sec for annealing/extension. The $C_t$ values of the genes were normalised to GAPDH. Finally, the mRNA expression of treated groups was normalised to control and relative expression was calculated using $2^{-\Delta\Delta Ct}$. The primer efficiency was assessed using a standard curve and melt curve analysis.

## Immunoblotting

The aggrecan, COL 2 and NF-kB expression levels were assessed using immunoblotting. The chondrocytes were treated as described in the previous section. After treatment, the medium was removed and cells were washed two times with cold PBS. Then, the cells were lysed in radioimmunoprecipitation assay (RIPA) (Nakalai Tesque, Japan). The plates containing lysis buffer were kept on ice for 15 min. Thereafter, the lysate was collected using a scraper, transferred into a microcentrifuge tube and centrifuged at 14,000 g for 15 min at 4°C. The supernatant was collected for protein estimation using a bicinchoninic acid assay kit (Thermo Fisher Scientific, United States). Then, 10 μg of protein was prepared in a sample buffer (Nacalai Tesque, Japan) containing 5% of 2-mercaptoethanol and denatured for 5 min at 95°C. The samples were separated on 8% SDS-polyacrylamide gel electrophoresis and transferred onto polyvinylidene difluoride membrane (Roche, Germany). Membranes were then incubated in a blocking buffer [2% BSA in Tris-buffered saline containing 0.1% Tween 20 (TBST)] with gentle agitation for 1 h. Subsequently, the membranes were washed three times for 10 min with TBST and probed with aggrecan monoclonal antibody (1:1000; GeneTex, USA), COL 2 polyclonal antibody (1:1000; GeneTex, USA), NF-kB p105/p50 polyclonal antibody (1:1000; Cell Signalling Technology, United States) and β-actin monoclonal antibody (1:1000; Santa Cruz Biotechnology, United States) at 4°C overnight with gentle agitation. After overnight incubation, the membranes were washed with TBST and incubated with goat anti-mouse-horseradish peroxidase or goat anti-rabbit-horseradish peroxidase (1:5000/10000; both from Thermo

Fisher Scientific, United States) for 1 h at room temperature. After washing, the protein bands were visualised using chemiluminescence detection kit (Chemi-Lumi One series for HRP; Nacalai Tesque, Japan) and Gel DOC system (BioRad gel Doc). The signal intensity of protein bands was normalised to β-actin and measured semi-quantitatively using Image J 1.51k software.

### Data analysis

Data are presented as mean ± SD. SPSS (version 25) was used for data analysis. A normality test was performed using the Shapiro-Wilk test. One-way ANOVA was used for data analysis, followed by Tukey multiple comparison tests. The standard curves for ELISA kits data were generated using a four-parameter logistic (4-PL) model with MyAssays software. The significant value was set at $p < 0.05$.

## Results

### Trilineage differentiation of SHED

The chondrogenic differentiation showed deposition of proteoglycan as evidenced by strong positive alcian blue staining (Fig 1A). The control group did not show strong positive staining to alcian blue (Fig 1B). For osteogenic differentiation, Alizarin Red S staining of the differentiated cells formed a bright red colour, indicating the presence of calcified nodules (Fig 1C), which were not observed in the control (Fig 1D). Adipogenic differentiation of the cells indicated the presence of intracellular lipid droplets as demonstrated by the Red Oil O staining (Fig 1E). There was no positive staining in the control (Fig 1F).

### SHED maintained MSC surface protein during the preparation of CM

To assess whether MSC surface proteins were not altered during CM preparation, the cells were analysed by flow cytometry. The results showed that the cells were positive for CD44, CD73, CD90 and CD105, but were negative for CD45, CD11b, CD19, CD34 and HLA-DR (Table 2). The cells cultured in SFM and CCM indicate no significant difference in the protein surface markers, suggesting SHED had maintained their MSC phenotype during the preparation of CM. The flow cytometry plots are depicted in S1 File.

### SHED expressed pluripotency markers with differential subcellular localisation

The expression levels of OCT4, NANOG and SOX2 were evaluated to confirm whether SHED that were serum-starved for the preparation of CM could have effects on the self-renewal and pluripotency of the cells. The results indicated that SHED cultured in CCM and SFM medium displayed the expression of these three transcription factors with differential subcellular localisation. OCT4 was localised to the cytoplasm (Fig 2A–2C), whereas NANOG was mainly localised to the nucleus and perinuclear (Fig 2D–2F). SOX2 was localised to both the nucleus and cytoplasm (Fig 2G–2I). Furthermore, the expression patterns were similar in the three culture conditions.

### CM contained secreted factors

The results showed that the amounts of TGF-β1 (Fig 3A) and IL-6 (Fig 3C) were significantly higher in S72 when compared with S48, suggesting the release of these factors were time-dependent. However, the release of IL-10 (Fig 3B) was highly reduced in S72 compared to S48.

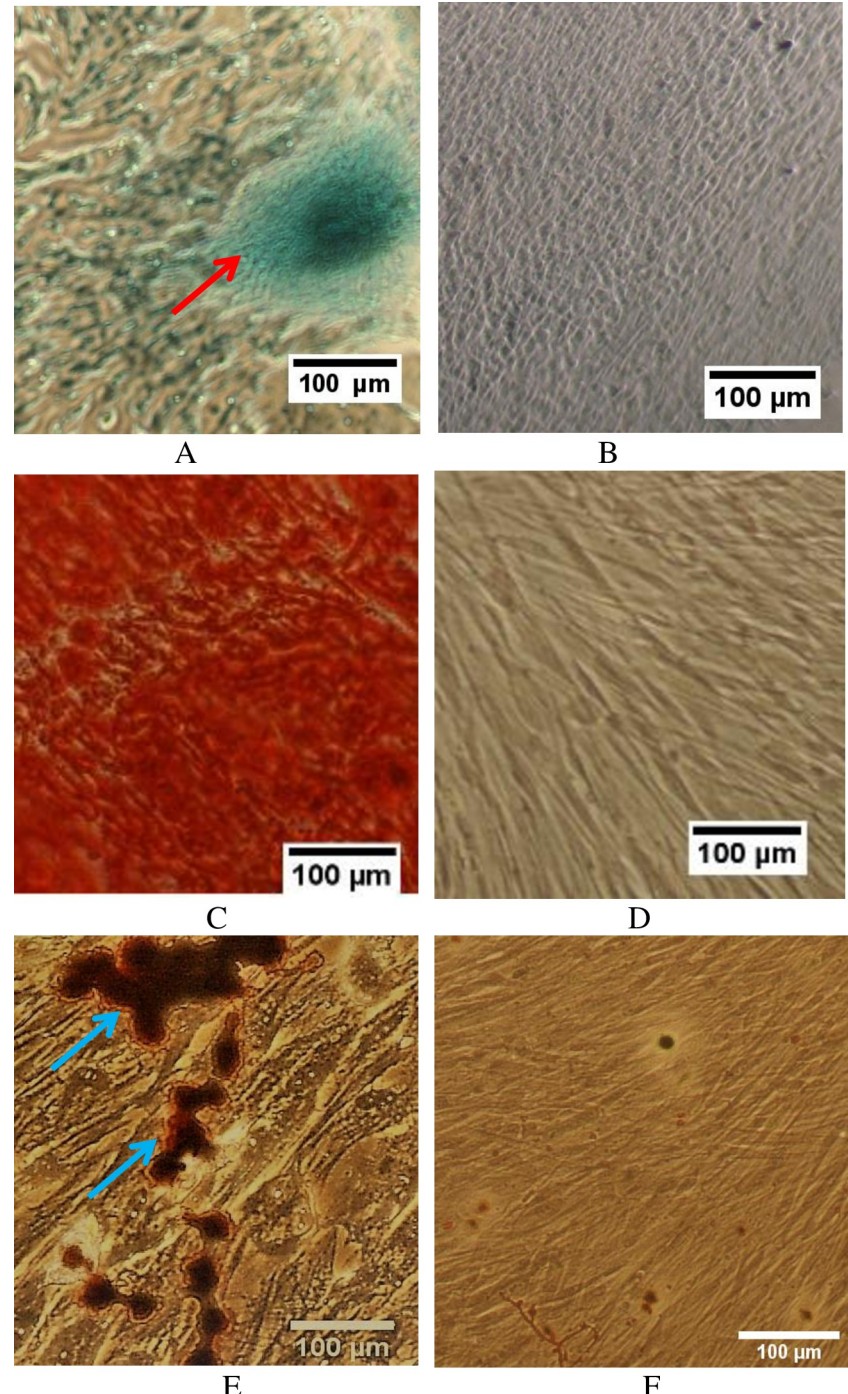

**Fig 1. Trilineage differentiation of SHED.** Differentiated SHED stained positive to alcian blue as shown by the formation of a spheroid (arrow) (A), whereas the control was negative to the staining (B). SHED showed the formation of calcified nodules for osteogenic differentiation (C) but not in the control (D). Lipid droplets were observed in adipogenic-differentiated cells as indicated by arrows (E). Control was negative to Red Oil O stain (F). Scale bar: 100 μm.

**Table 2. Expression of MSC phenotype in SHED cultured under different conditions.**

| Group | CD44 (%) | CD73 (%) | CD90 (%) | CD105 (%) | CD11b, CD19, CD34, CD45, HLA-DR (%) |
|---|---|---|---|---|---|
| CCM | 99.50±0.70 | 98.85±1.63 | 98.80±0.29 | 99.70±0.28 | 0.75±0.49 |
| SFM (48 h) | 99.90±0.14 | 99.47±0.15 | 99.32±0.26 | 99.82±0.09 | 2.10±2.26 |
| SFM (72 h) | 97.70±2.97 | 99.37±0.20 | 98.65±0.70 | 99.75±0.20 | 2.55±2.89 |

Values are mean ± SD, n = 4 from two independent experiments. Data are not significantlydifferent between the groups using one-way ANOVA. CCM-complete culture medium, SFM (48 h)–incubated in serum-free medium for 48 h, SFM (72 h)- incubated in serum-free medium for 72 h, CD-cluster of differentiation, HLA-DR-human leukocyte antigen-antigen D related.

## CM improved the viability of OA chondrocytes

The effect of CM was evaluated on the viability of stimulated chondrocytes at different time points (Fig 4). At 24 h, the percentage viability in stimulated chondrocytes treated with CM was significantly higher compared to SFM control. There was no significant difference in the viability of cells treated with CM when compared with non-stimulated cells incubated in CCM. At 48 h, the viability of cells incubated with CM was similar compared to CCM. The lowest percentage of viability was observed in SFM control. When the incubation period was increased to 72 h, the viability of the cells treated with S48 was significantly higher compared to SFM or SFM supplemented with TGF- β1. The viability of cells treated with CM was not

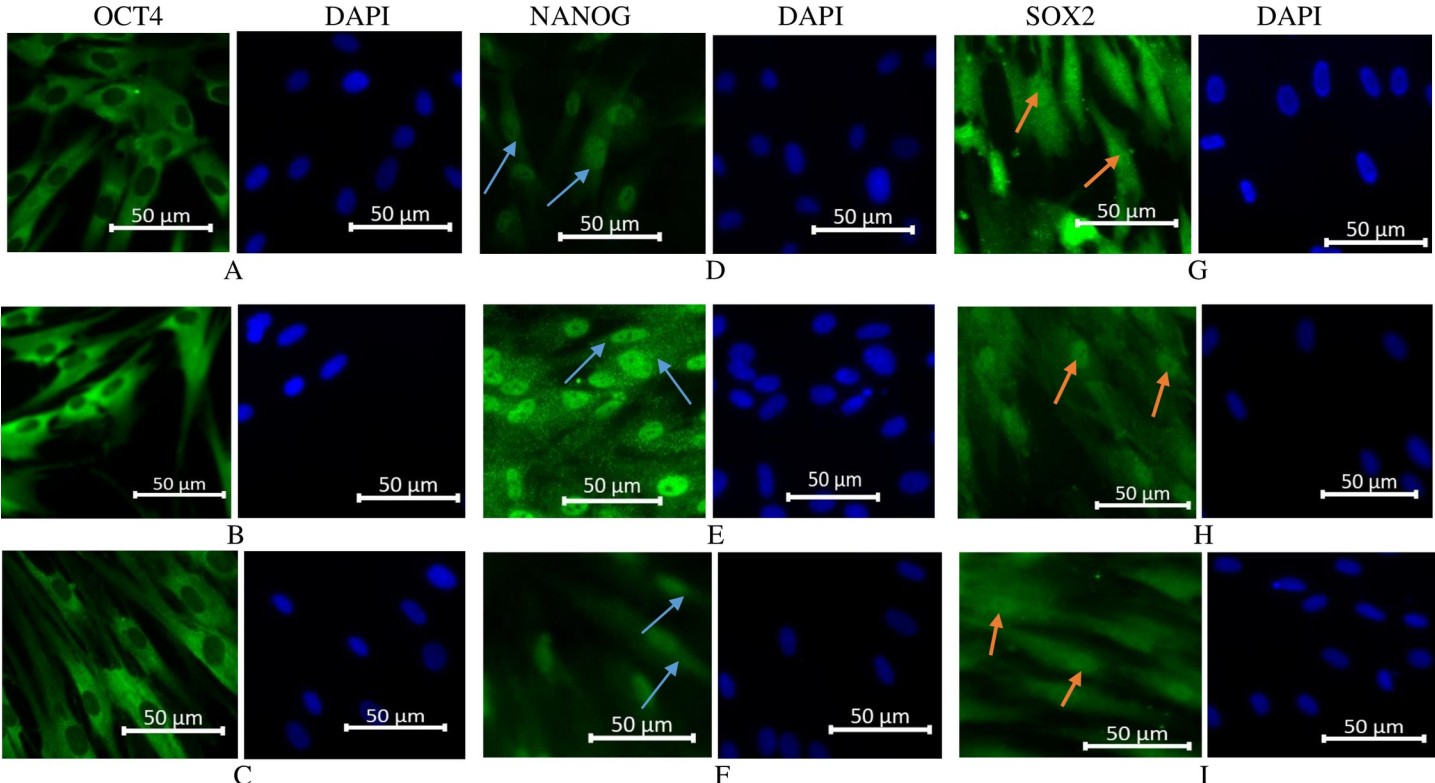

**Fig 2. Expression of pluripotency markers in SHED cultured under different conditions.** Expression of OCT4 was localised to the cytoplasm (A-C), whereas expression of NANOG was localised to the nucleus and perinuclear (indicated by arrows) (D-F). SOX2 was localised to nucleus and cytoplasm (G-I). Arrows indicate the nucleus expression of SOX2. Cultivated in CCM (A, D & G), cultivated in SFM for 48 h (B, E & H), cultivated in SFM for 72 h (C, F & I). CCM-complete culture medium, SFM-serum-free medium. Scale bar: 50 μm.

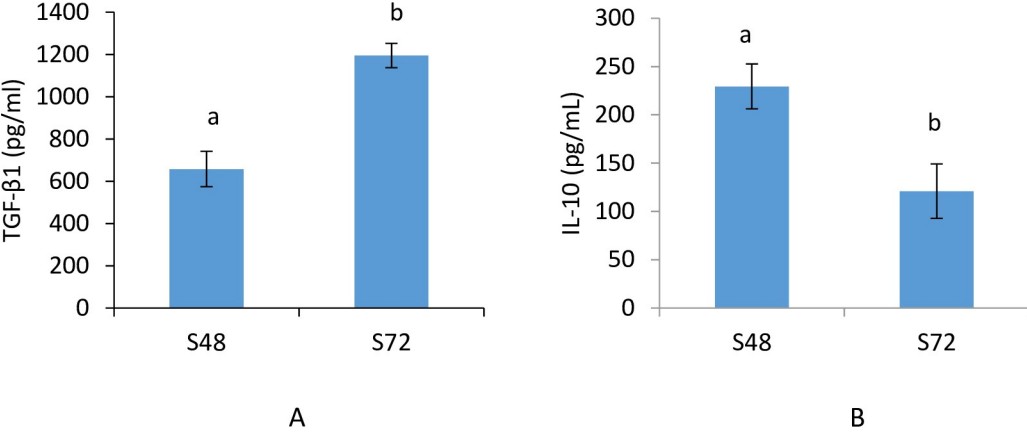

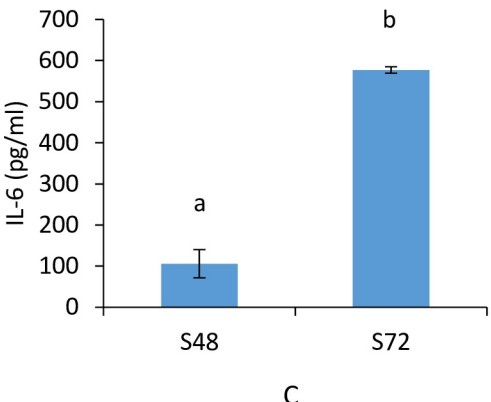

**Fig 3. Level of secreted factors in CM.** (A) TGF- β1 (B) IL-10 (C) IL-6. Values are mean ± SD, n = 3. Bars with different letter are significantly different using one-way ANOVA, followed by Tukey post hoc tests. S48—CM collected at 48 h of incubation, S72- CM collected at 72 h of incubation.

significantly different when compared with CCM. Overall, the percentage of viable cells in CM groups were greater than 88% for the three time points.

## CM decreased the level of MMP-13 and enhanced anti-inflammation in OA chondrocytes

To minimize the impact of cell viability on the results, the incubation time of subsequent assays were evaluated at 48 h because the viability of the cells in SFM and SFM supplemented with TGF-β1 groups were below 80% at 72 h. The stimulation of chondrocytes with IL-1β significantly increase the MMP-13 level (Fig 5A) as evidenced by the highest level observed in the SFM control. However, treatment with CM mitigates the increased level of MMP-13.

The level of IL-6, TGF-β1 and IL-10 were evaluated after the stimulated chondrocytes were treated with CM (Fig 5B–5D). When chondrocytes were stimulated with IL-1β and incubated in SFM, the level of IL-6 was significantly elevated (18214.0 ± 186.5 pg/mL). However, incubation with CM significantly reduced the level of IL-6 compared to SFM control (Fig 5B). The level of TGF-β1 in non-stimulated chondrocytes incubated with CCM was similar to the stimulated cells incubated in CM (Fig 5C). TGF-β1 level was higher in CM compared to SFM

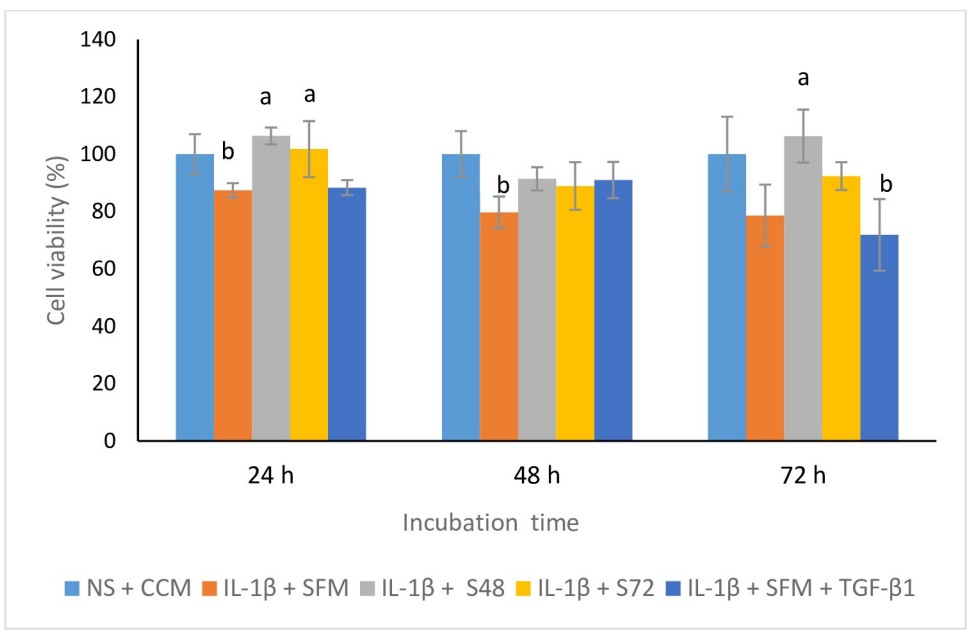

**Fig 4. The percentage viability of stimulated chondrocytes incubated with CM for three time points.** Values are mean ± SD, n = 4, [a]$p<0.05$ when compared with IL-1β + SFM & [b]$p<0.05$ when compared with IL-1β + CCM using one-way ANOVA, followed by Tukey post hoc tests. TGF-β1–transforming growth factor-β1, IL-1β -interleukin-1β, NS-non-stimulated, CCM-complete culture medium, SFM- serum-free medium, S48- CM collected at 48 h of incubation, S72-CM collected at 72 h of incubation.

group. The level of TGF-β1 was better in stimulated chondrocytes treated with CM compared to that of SFM supplemented with TGF-β1.

IL-10 was significantly elevated in stimulated chondrocytes incubated with SFM compared to non-stimulated cells incubated in CCM (Fig 5D). When the stimulated chondrocytes were cultured in CM, the level of IL-10 was significantly reduced compared to the level observed in stimulated cells incubated with SFM.

## CM downregulates the mRNA expression of MMP-13 and increases the expression of aggrecan and COL 2 by downregulating NF-kB

IL-1β downregulates the expression level of aggrecan as shown by decreased expression in stimulated cells compared to non-stimulated cells incubated in CCM (Fig 6A). However, when the cells were treated with CM the aggrecan expression was increased compared to SFM control.

COL 2 mRNA expression in stimulated chondrocytes (Fig 6B) indicates that stimulation of chondrocytes with IL-1β downregulates COL 2 expression. When the stimulated cells were treated with CM, there was an increase in the expression of COL 2 compared to SFM control.

The results (Fig 6C) showed that incubation of the stimulated chondrocytes with CM significantly downregulates the mRNA expression of MMP-13 compared to SFM or SFM supplemented with TGF-β1. Interestingly, the MMP-13 expression was similar in CM groups when compared with non-stimulated cells incubated in CCM.

Since the activation of many inflammatory mediators and catabolic molecules are mediated by NF-kB, we investigated whether CM could modulate its expression. The mRNA expression of NF-kB (Fig 6D) was highly upregulated in stimulated chondrocytes compared to non-stimulated cells incubated with CCM, suggesting the role of IL-1β in NF-kB activation. When the

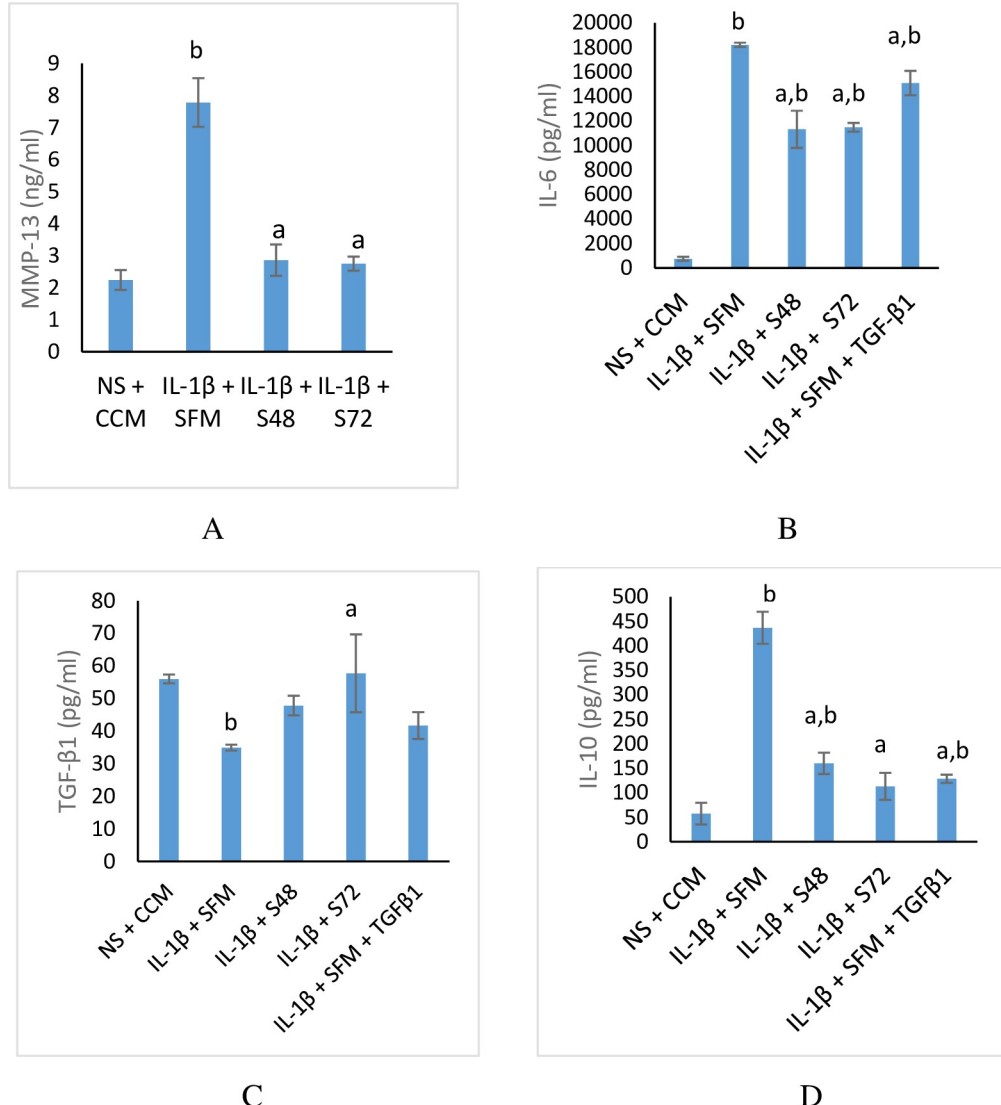

**Fig 5. MMP-13 and cytokine level in osteoarthritic chondrocytes incubated with CM.** Values are mean ± SD, n = 3. [a]p<0.05 when compared with IL-1β + SFM & [b]p<0.05 when compared with NS + CCM using one-way ANOVA, followed by Tukey post hoc multiple comparison tests. MMP-13-matrix metalloproteinase-13, IL-1β-interleukin-1β, IL-6- interleukin-6, IL-10- interleukin-10, TGF-β1-transforming growth factor-β1, NS-non- stimulated, CCM-complete culture medium, SFM-serum- free medium, S48–CM collected at 48 h of incubation, S72-CM collected at 72 h of incubation.

cells were incubated with CM, the mRNA expression of NF-kB was significantly downregulated compared to SFM control.

## CM increased the expression of matrix proteins and decreased NF-kB in OA chondrocytes

The protein expression of aggrecan, COL 2 and NF-kB was investigated by immunoblotting to validate the results of gene expression (Fig 7A–7D). Immunoreactive bands of the various protein assessed are depicted in Fig 7A. The stimulation of chondrocytes with IL-1β downregulates aggrecan and COL 2 and upregulates NF-kB expression. However, treatment with CM mitigates the effect of IL-1β. The expression of aggrecan was significantly upregulated in

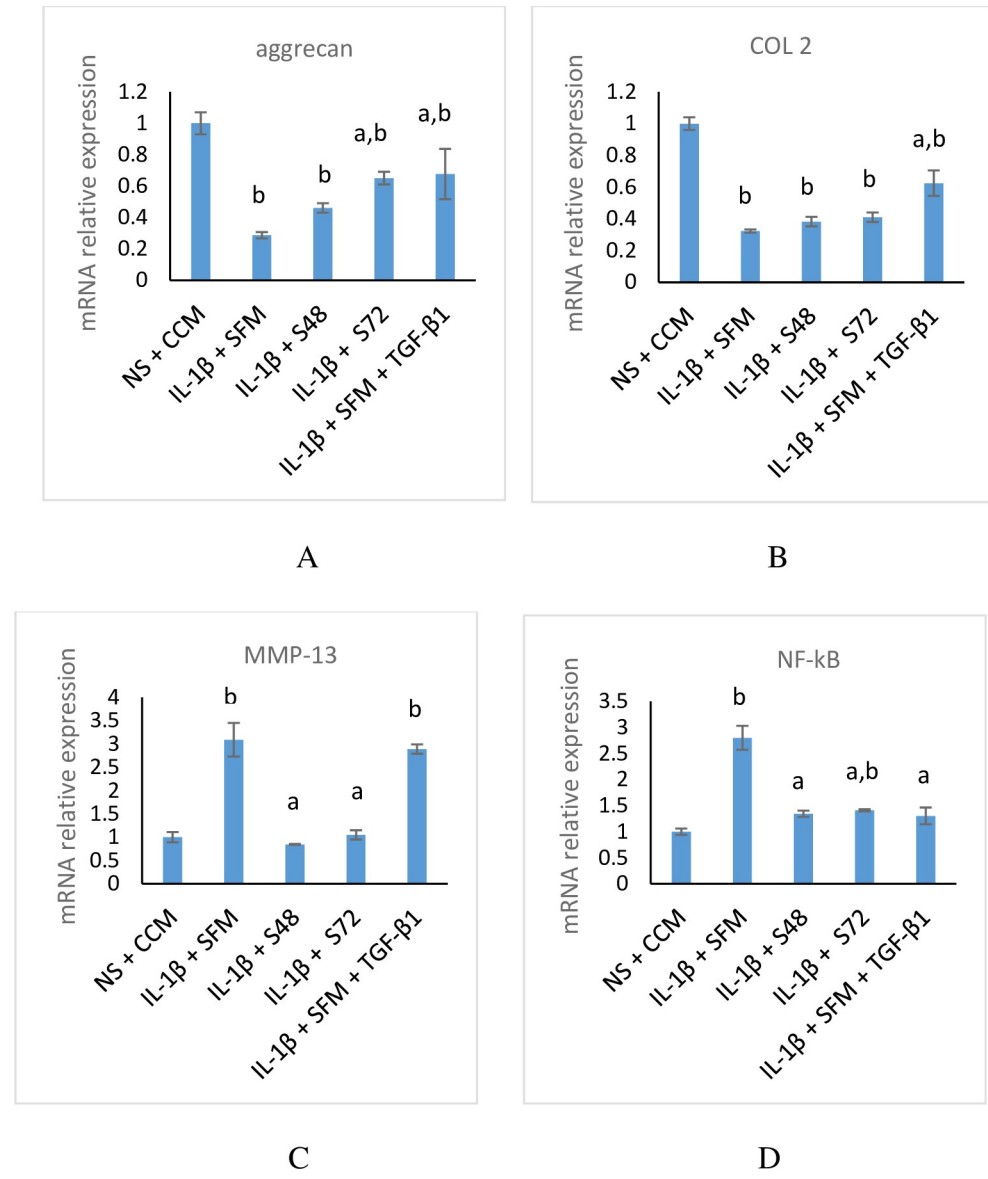

**Fig 6. mRNA expression of matrix genes and NF-kB in stimulated chondrocytes incubated with CM.** Data are mean ± SD, n = 3. (A) aggrecan (B) COL 2 (C) MMP-13 (D) NF-kB. Expression is relative to NS + CCM control after 48 h of incubation. [a]$p$<0.05 when compared with IL-1β + SFM & [b]$p$< 0.05 when compared with NS + CCM using one-way ANOVA, followed by Tukey post hoc test. NS-non-stimulated, SFM- serum-free medium, CCM-complete culture medium, IL-1β-interleukin-1β, TGF-β1-transforming growth factor-β1, COL 2-collagen type 2, NF-kB-nuclear factor-kappa B, MMP-13- matrix metalloproteinase-13, S48- CM collected at 48 h of incubation, S72- CM collected at 72 h of incubation.

stimulated chondrocytes incubated with CM compared to SFM control (Fig 7B). Relative COL 2 expression (Fig 7C) showed that treatment of stimulated chondrocytes with CM ameliorates the catabolic activity of IL-1β by increasing its expression compared to SFM control. A similar pattern of the result was observed when the CM treated groups were compared with SFM supplemented with TGF-β1 group.

The expression of NF-kB transcription factor (Fig 7D) was significantly downregulated in stimulated cells incubated with CM compared to SFM control. The expression of NF-kB

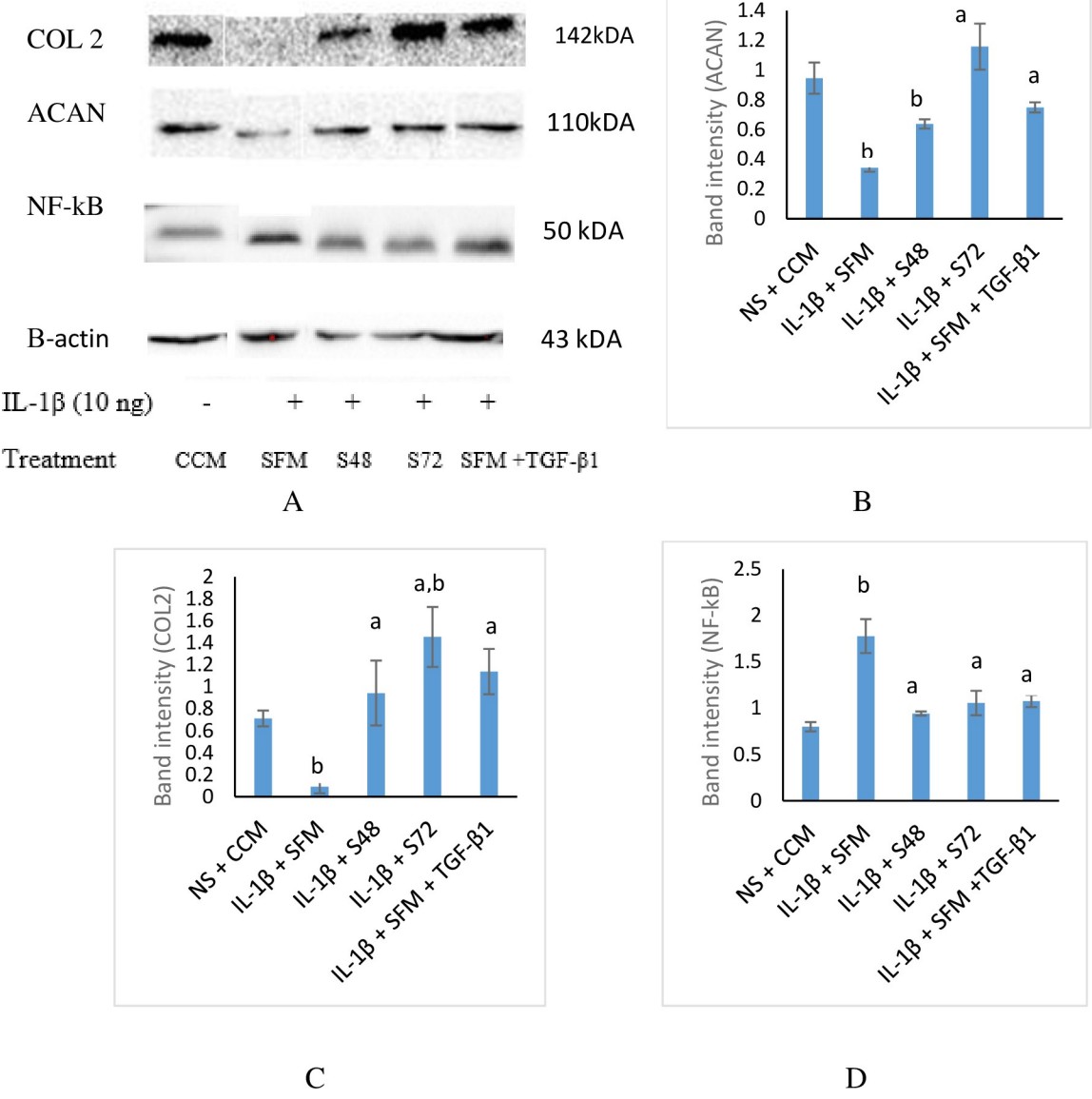

**Fig 7. Protein expression of aggrecan, COL 2 and NF-kB in stimulated chondrocytes treated with CM.** Data are mean ± SD, n = 3. (A) Immunoreactive bands of various protein. Relative band intensity of ACAN (B), COL 2 (C) and NF-kB (D) normalised to β-actin. [a]$p < 0.05$ when compared with IL-1β + SFM & [b]$p < 0.05$ when compared with NS + CCM using one-way ANOVA, followed by Tukey post hoc test. ACAN- aggrecan, COL 2-collagen type 2, NF-kB-nuclear factor-kappa B, CCM-complete culture medium, NS-non-stimulated, IL-1β-interleukin 1β, TGF- β1-transforming growth factor β1, S48 –CM collected at 48 h of incubation, S72- CM collected at 72 h of incubation, '+'- plus IL-1β, '-'–no IL-1β.

observed in the stimulated cells incubated with CM was not significantly different when compared with non-stimulated chondrocytes incubated in CCM or stimulated incubated in SFM supplemented with TGF-β1.

## Discussion

Recent studies have demonstrated that stem cells secrete bioactive molecules that are important for tissue repair and regeneration [14, 20, 21]. In this study, SHED were characterised by the expression of MSC and pluripotency markers. The expression of matrix proteins, MMP-13

and the level of IL-6, IL-10 and TGF-β1 markers were evaluated in stimulated chondrocytes treated with CM. NF-kB expression was also investigated to delineate how the CM could modulate this transcription factor in OA chondrocytes.

SHED were differentiated to osteocytes, chondrocytes and adipocytes, confirming the cells were MSCs. To assess the possibility of phenotypic alterations or spontaneous differentiation, the MSC surface proteins were investigated after the collection of CM. It has been previously reported that *in vitro* expansion of MSCs may undergo genetic and/or epigenetic alterations and spontaneous osteogenic differentiation [22, 23]. The result showed that SHED expressed MSC positive markers and were negative for haematopoietic markers. The expressions of MSC phenotype were similar in SHED cultured in CCM and SFM. This shows that SHED pre-conditioned for the preparation of CM did not cause any significant phenotypic alterations. It has been demonstrated that MSC phenotypic markers are involved in the regulation of cellular proliferation, migration, differentiation and immune response [24, 25].

The expression of NANOG, OCT4 and SOX2 were detected in SHED pre-conditioned for CM production and SHED cultured in CCM with differential subcellular localisation. The expression pattern of these transcription factors was similar for SHED culture under different conditions. OCT4 was localised to the cytoplasm in this study. However, differential subcellular localisation of OCT4 has been previously demonstrated. Oka et al. [26] demonstrated that OCT4 is a nucleocytoplasmic shuttling protein which is capable of maintaining self-renewal of embryonic stem cells. Furthermore, the expression of OCT4 in human dental stem cells was detected in the nucleus at the early passages with weak cytoplasmic staining but was localised to the cytoplasm at the late passage [27]. These transcription factors are the important master regulator for the pluripotency and self-renewal. A report indicates that NANOG and OCT4 act cooperatively to regulate MSC proliferation and differentiation [28].

The use of stem cell secretome as therapeutic strategies may easy banking and standardisation of interventions for clinical applications. The results showed variations in the levels of TGF-β1, IL-10 and IL-6. The detection of growth factors, cytokines and chemokines in CM has been previously reported [29]. TGF-β have been reported to stimulate ECM synthesis and decrease matrix protein degradation in chondrocytes [30]. IL-6 is a cytokine with both pro-inflammatory and anti-inflammatory effects, whereas IL-10 is an anti-inflammatory cytokine. The CM was able to support chondrocyte viability. This observed effect may be attributed to the presence of growth factors in the CM that stimulates chondrocytes proliferation and growth.

Due to the rapid response of chondrocytes to cytokine stimulation has made it the most widely used model for *in vitro* osteoarthritis [31]. The present study demonstrated an increased level of MMP-13 and IL-6 in the response of chondrocytes to IL-1β stimulation. Furthermore, increase IL-6 and MMP-13 level have been reported in osteoarthritis [32, 33]. CM significantly decreased the levels of MMP-13 and IL-6, whereas an increased level of TGF-β1 was observed. IL-6 has been shown to inhibits COL 2 synthesis in chondrocytes [34]. MMP-13 acts to degrade collagen type 2 and proteoglycans in OA [35]. The decreased level of MMP-13 suggests the beneficial role of CM. This finding is in agreement with previous studies on the role of CM in decreasing MMP-13 [36, 37]. The articular cartilage matrix turnover and homeostasis are dependent on TGF-β-mediated anabolic signalling that is crucial for the maintenance of articular cartilage [38]. TGF-β can stimulate the synthesis of aggrecan and COL 2 and inhibits the degradation of these proteins by increasing the synthesis of tissue inhibitor of metalloproteinase [38, 39]. The increase TGF-β1 level observed suggests that the CM is capable of stimulating ECM synthesis. The level of IL-10 was higher in SFM control compared to CM treated groups or non-stimulated treated with CCM. Studies have shown an increase expression of IL-10 in OA patient [40]. However, the chondroprotective role of IL-10

has been previously demonstrated [41]. IL-10 acts through activation of the bone morphogenetic protein signalling pathway to protects chondrocytes [42]. The decreased level of IL-10 observed in the CM groups may be attributed to the presence of IL-10 in the CM, which was able to mitigate the effect of IL-1β without it being secreted by the cells. Moreover, the increased level of IL-10 observed in SFM control could be attributed to the fact that the cells were induced to secrete this anti-inflammatory cytokine to mitigate the effect of IL-1β assault. This shows that the CM protected chondrocytes from synthesizing IL-10 in response to IL-1β stimulation, suggesting the anti-inflammatory activity of the CM.

The results of the gene expression of COL 2, aggrecan, MMP-13 and NF-kB were investigated. The expression of COL 2 and aggrecan was upregulated, whereas MMP-13 expression was downregulated following CM intervention. The data demonstrated that CM mitigates the upregulation of MMP-13 expression following stimulation with IL-1β. The findings suggest the anti-degradative effect of CM. Increased expression of COL 2 and aggrecan by CM suggests the regenerative activity of CM.

The activation of the NF-kB pathway is triggered by phosphorylation and degradation of IkB proteins, which facilitates the expression of inflammatory mediators and matrix degradative molecules [43]. In OA chondrocytes, activation of canonical NF-kB signalling induces the expression of IL-1β, TNF-α and MMPs [44, 45]. In this study, CM was able to downregulate mRNA expression of NF-kB. This suggests that downregulation of MMP-13 is mediated through inhibition of NF-kB transcription factor. Consistent with this observation, some studies have shown similar results on osteoarthritic chondrocytes [46, 47].

Furthermore, protein expression of aggrecan, COL 2 and NF-kB were investigated to validate the result of gene expression. CM increases the expression of aggrecan and COL 2, which were downregulated following IL-1β stimulation. The protein expression also showed that CM reduced NF-kB expression. Taken together, the results of protein expression corroborated that of gene expression. The data presented support the therapeutic benefits of CM in cartilage repair and regeneration.

## Conclusions

The present study provides an understanding of the therapeutic benefit of CM for OA. SHED showed the expression of MSC phenotype and pluripotent markers. The production of CM under serum starvation did not cause changes to the expression of MSC phenotype and pluripotent markers and the latter displayed dynamics in subcellular localisation.

It is evidenced that both CM protected chondrocytes by increasing matrix proteins and suppresses MMP-13 expression. The CM also mitigates the inflammatory assault induced by IL-1β. The regenerative effect of CM could be attributed to secreted factors that modulate the catabolic processes toward anabolic phenotype through downregulation of NF-kB. Together, the study demonstrates the potential of cell-free based therapy for cartilage repair and regeneration.

## Supporting information

**S1 File.**
(DOCX)

## Acknowledgments

This research work was supported by Putra Grant, Universiti Putra Malaysia with grant number: GPB/9657800.

## Author Contributions

**Conceptualization:** Suleiman Alhaji Muhammad, Noor Hayaty Abu Kasim, Sharida Fakurazi.

**Formal analysis:** Suleiman Alhaji Muhammad, Norshariza Nordin, Muhammad Zulfadli Mehat, Sharida Fakurazi.

**Funding acquisition:** Suleiman Alhaji Muhammad, Norshariza Nordin, Paisal Hussin, Sharida Fakurazi.

**Investigation:** Suleiman Alhaji Muhammad, Paisal Hussin.

**Writing – original draft:** Suleiman Alhaji Muhammad.

**Writing – review & editing:** Norshariza Nordin, Paisal Hussin, Muhammad Zulfadli Mehat, Noor Hayaty Abu Kasim, Sharida Fakurazi.

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
