## [Decision Letter · Decision Letter 0]

24 Jul 2020

PONE-D-20-20979

Protective effects of stem cells from human exfoliated deciduous teeth derived conditioned medium on osteoarthritic chondrocytes

PLOS ONE

Dear Dr. Muhammad,

Thank you for submitting your manuscript to PLOS ONE. After careful consideration, we feel that it has merit but does not fully meet PLOS ONE’s publication criteria as it currently stands. Therefore, we invite you to submit a revised version of the manuscript that addresses the points raised during the review process.

This paper is of interest pending some minor amendments that have been specified by each of the two referees. Authors must follow all their concerns and amend figures and text in red in their revised text. 

We look forward to receiving your revised manuscript.

Kind regards,

Gianpaolo Papaccio, M.D., Ph.D.

Academic Editor

PLOS ONE

Journal Requirements:

Reviewers' comments:

Reviewer's Responses to Questions

**Comments to the Author**

1. Is the manuscript technically sound, and do the data support the conclusions?

Reviewer #1: Yes

Reviewer #2: Yes

2. Has the statistical analysis been performed appropriately and rigorously? 

Reviewer #1: Yes

Reviewer #2: Yes

3. Have the authors made all data underlying the findings in their manuscript fully available?

Reviewer #1: Yes

Reviewer #2: Yes

4. Is the manuscript presented in an intelligible fashion and written in standard English?

Reviewer #1: Yes

Reviewer #2: No

5. Review Comments to the Author

Reviewer #1: In this paper Authors evaluated the regenerative effect of stem cells from human exfoliated deciduous teeth (SHED) conditioned medium (CM) on OA chondrocytes.

The paper is interesting and well organized and the experiments are well performed. Some minor changes are needed.

English language should be revised.

In figure 1 images should be presented to a better resolution.

Authors should adjust histograms: they should have the same layout (bold and normal characters, border…)

Reviewer #2: This study was designed to assess how conditioned medium (CM)

from SHED could modulate tissue microenvironment of catabolic processes in OA chondrocytes. The mansucript is interesting, but there are some points that need to be addressed.

MMP13 must be evaluated by western blotting. Table 2 must be deleted or added as supplemental file and the data must be reported as histogram.

In figure 2, OCT4 is distributed in cytoplasm. Usually, in stem cell, it is nuclear. Clarify this point.

The English language must be revised by native English speaker.

6. PLOS authors have the option to publish the peer review history of their article (what does this mean?). If published, this will include your full peer review and any attached files.

Reviewer #1: No

Reviewer #2: No

---

## [Author Response · Author response to Decision Letter 0]

1 Aug 2020

Academic Editor

Comment

Thank you for submitting your manuscript to PLOS ONE. After careful consideration, we feel that it has merit but does not fully meet PLOS ONE’s publication criteria as it currently stands. Therefore, we invite you to submit a revised version of the manuscript that addresses the points raised during the review process.

This paper is of interest pending some minor amendments that have been specified by each of the two referees. Authors must follow all their concerns and amend figures and text in red in their revised text. 

Response

Thank you for finding merit in our manuscript. The concerns raised by the referees are addressed in the revised version as recommended.

Reviewer #1: 

In this paper Authors evaluated the regenerative effect of stem cells from human exfoliated deciduous teeth (SHED) conditioned medium (CM) on OA chondrocytes.

The paper is interesting and well organized and the experiments are well performed. Some minor changes are needed.

Response

Thank you for the commendation and observations made which aimed to improve the paper.

Comment 

English language should be revised.

Response 

English language has been thoroughly checked and improved as suggested.

Comment 

In figure 1 images should be presented to a better resolution.

Response 

Figure 1 resolution was improved as recommended.

Comment 

Authors should adjust histograms: they should have the same layout (bold and normal characters, border…)

Response

Histograms have been adjusted as suggested in the revised version

Reviewer #2: 

This study was designed to assess how conditioned medium (CM)

from SHED could modulate tissue microenvironment of catabolic processes in OA chondrocytes. The manuscript is interesting, but there are some points that need to be addressed.

Response 

Thank you for the observations aimed at improving our manuscript.

Comment 

MMP13 must be evaluated by western blotting. 

Response 

Western blotting of MMP13 was not reported in the current study. However, MMP-13 level was assessed using ELISA after treatment of OA chondrocytes with CM and was earlier presented in Fig 5a.

 Comment 

Table 2 must be deleted or added as supplemental file and the data must be reported as histogram.

Response

The reason we excluded the histogram for Table 2 was to avoid the results being presented twice since they were derived from the histogram. There are three groups of different surface proteins, which would make the results awkward for the readers if presented in histograms. Based on this, we have retained the results presented in Table 2 and provided the histogram in the supplementary file 1.

Comment 

In figure 2, OCT4 is distributed in cytoplasm. Usually, in stem cell, it is nuclear. Clarify this point.

Response

The explanation on the localization of OCT4 in the cytoplasm is provided in the revised version with supporting literature. OCT4 is a nucleocytoplasmic shuttling protein as previously reported. 

Comment 

The English language must be revised by native English speaker.

Response

The English language has been extensively revised as suggested in the amended version.

---

## [Decision Letter · Decision Letter 1]

18 Aug 2020

Protective effects of stem cells from human exfoliated deciduous teeth derived conditioned medium on osteoarthritic chondrocytes

PONE-D-20-20979R1

Dear Dr. Muhammad,

We’re pleased to inform you that your manuscript has been judged scientifically suitable for publication and will be formally accepted for publication once it meets all outstanding technical requirements.

Kind regards,

Gianpaolo Papaccio, M.D., Ph.D.

Academic Editor

PLOS ONE

Additional Editor Comments (optional):

Reviewers' comments:

Reviewer's Responses to Questions

**Comments to the Author**

1. If the authors have adequately addressed your comments raised in a previous round of review and you feel that this manuscript is now acceptable for publication, you may indicate that here to bypass the “Comments to the Author” section, enter your conflict of interest statement in the “Confidential to Editor” section, and submit your "Accept" recommendation.

Reviewer #1: All comments have been addressed

Reviewer #2: All comments have been addressed

2. Is the manuscript technically sound, and do the data support the conclusions?

Reviewer #1: (No Response)

Reviewer #2: Yes

3. Has the statistical analysis been performed appropriately and rigorously? 

Reviewer #1: (No Response)

Reviewer #2: Yes

4. Have the authors made all data underlying the findings in their manuscript fully available?

Reviewer #1: (No Response)

Reviewer #2: Yes

5. Is the manuscript presented in an intelligible fashion and written in standard English?

Reviewer #1: (No Response)

Reviewer #2: Yes

6. Review Comments to the Author

Reviewer #1: (No Response)

Reviewer #2: (No Response)

7. PLOS authors have the option to publish the peer review history of their article (what does this mean?). If published, this will include your full peer review and any attached files.

Reviewer #1: No

Reviewer #2: No

---

## [Editor Report · Acceptance letter]

27 Aug 2020

PONE-D-20-20979R1 

Protective effects of stem cells from human exfoliated deciduous teeth derived conditioned medium on osteoarthritic chondrocytes 

Dear Dr. Muhammad:

I'm pleased to inform you that your manuscript has been deemed suitable for publication in PLOS ONE. Congratulations! Your manuscript is now with our production department. 

Kind regards, 

on behalf of

Prof. Gianpaolo Papaccio 

Academic Editor

PLOS ONE